

# Uncertainty in Land Carbon Fluxes Simulated by CMIP6 Models from Treatments of Crop Distributions and Photosynthetic Pathways

Joseph Ovwemuvwose[1], Ian C. Prentice[2,3], and Heather. D. Graven[1]

[1]Department of Physics, Imperial College London, London, UK.
[2]Georgina Mace Centre for the Living Planet, Department of Life Sciences, Imperial College London, London, UK
[3]Department of Earth System Science, Tsinghua University, Beijing, China

*Correspondence to*: Joseph Ovwemuvwose (j.ovwemuvwose22@imperial.ac.uk & josovw@gmail.com)

**Abstract.** A reliable representation of the diversity of vegetation in terrestrial ecosystems is needed for the accurate simulation of present and future biogeochemical cycling and global climate, particularly as climate change affect different vegetation types differently. We compare the distributions of crops and of $C_3$ versus $C_4$ photosynthetic pathways in both natural vegetation

and crops across Earth System Models in the 6th Coupled Model Intercomparison Project (CMIP6). We find a large range in vegetation type for area, gross primary production (GPP) and carbon stock change in both natural vegetation and croplands across the models. Even though 10 of 11 models used Land Use Harmonization (LUH2) crop areas as input data, modeled total crop area ranges from -28 to +10 % of the data-based estimate. The $C_3$ and $C_4$ crop areas were -56 to +15 % and -100 to +38 % of LUH2 for 2014, respectively. The $C_4$ fraction of total vegetation area in the models is 9-25 %, compared to 17 % in

observation-based estimates. Total global GPP varies by a factor of two across the models, and the $C_4$ fraction of GPP ranges from 12 to 27 %. Simulated trends in the fraction of GPP by $C_3$ versus $C_4$ vegetation type (-20 to +29 %) would have changed global isotopic discrimination by -0.35 to +0.11 ‰ over 1975-2005, indicating that modeled changes in vegetation type do not account for the +0.7 ‰ increase indicated by atmospheric data. Disparity in vegetation types contributes to uncertainty in land carbon fluxes and further constraints and improvements in models are needed.

## 1 Introduction


The terrestrial biosphere captures ~30 % of anthropogenically emitted $CO_2$ annually  (Friedlingstein et al., 2025) reducing $CO_2$ accumulation in the atmosphere and the accompanying global warming. However, this $CO_2$ uptake may be sensitive to future climate change (Arora et al., 2020). Understanding the mechanisms contributing to the $CO_2$ uptake, including the role of different types of vegetation and land use, is essential to understanding potential carbon-climate feedback and future changes

to the terrestrial carbon cycle.



The implementation of land use and land use change (LUC) and its impact on vegetation cover and dynamics are important components of terrestrial biosphere model development (Hu et al., 2021; Hurtt et al., 2020; Wang et al., 2022). LUC is mainly driven by agriculture. The fraction of global land area used for agriculture increased from 14 % in 1850 to about 37 % in 2015 (Hurtt et al., 2020) and LUC emitted about 118 PgC between 1850 and 2020 (Houghton & Castanho, 2023). In addition to

$CO_2$ emissions, the biophysical effects of land conversion also drive global temperature rise due to changes in surface albedo (Arora & Boer, 2010; Houghton et al., 2012). The alteration of the land surface will continue to be significant in the future as, for example, it has been projected that 14 % of vegetation and 5 % of soil carbon stocks will be lost to cropland expansion globally over 2010-2050 in a "middle-of-the-road" scenario (Molotoks et al., 2018a).

To represent plant diversity and function, most model developers use plant functional types (PFTs) that group vegetation by

similar features such as growth form, ecological requirements, and photosynthetic pathways. This helps to account for the variation in adaptive mechanisms and ecological distribution of different plants (Haxeltine & Prentice, 1996; Hurtt et al., 2020; Wullschleger et al., 2014). One important characteristic of plants is their use of either the $C_3$ or $C_4$ photosynthetic pathways. In $C_3$ plants the first product of the photosynthetic pathway is a three-carbon molecule called 3-phosphoglycerate (3-PGA), while in $C_4$ vegetation, it is a four-carbon molecule called oxaloacetate. $C_3$ and $C_4$ types also differ in their response to

changes in soil moisture content, temperature, $CO_2$, and light (Luo et al., 2024).

Under increasing temperature, especially increases above the thermal optimum threshold, productivity in $C_3$ vegetation is limited by an increase in photorespiration (Hermida-Carrera et al., 2016), which reduces photosynthetic efficiency in $C_3$ plants. In contrast, $C_4$ species overcome photorespiration through their $CO_2$ concentrating mechanism that increases the amount of $CO_2$ at the site of carboxylation (da Silva et al., 2020), leading to an abundance of $C_4$ species (natural grasses and crops such

as maize) in hot areas in the tropics and sub-tropics (Luo et al., 2024). While rising temperature generally favours $C_4$ plants, rising atmospheric $CO_2$ concentration confers a physiological advantage upon $C_3$ species (Polley et al., 1994) due to a reduction in photorespiration and increase in water use efficiency. This $CO_2$ fertilization effect is responsible for the increasing presence of $C_3$ woody species in previously $C_4$ dominated grasslands (Luo et al., 2024). Shifts in $C_3$ and $C_4$ species composition and carbon fluxes across different regions are projected to continue in future due to changing temperature, water availability and

increasing $CO_2$ concentration in the atmosphere (Cortés et al., 2021; Smith & Boers, 2023).

A change in the relative contributions of $C_3$ and $C_4$ vegetation to global productivity may contribute to a global trend in stable photosynthetic carbon isotope discrimination (Δ) because $C_3$ plants discriminate against carbon-13 more strongly than $C_4$ plants (Farquhar et al., 1989). Since atmospheric studies have indicated that Δ increased by 0.7 ‰ over 1975-2005 globally (Keeling et al 2017), and by 0.4 ‰ over 2000-2011 in the Northern Hemisphere (Peters et al 2018), understanding the effect

of changes in vegetation type on discrimination would help to quantify the environmental and physiological factors influencing plant function and resulting discrimination, including soil moisture content, vapour pressure deficit and stomatal conductance (Cornwell et al., 2018; Griffis et al., 2010).

Currently, the relative contribution, and its change over time, of $C_3$ vs $C_4$ vegetation to global terrestrial biosphere productivity and their ecological roles in climate change mitigation are not well-known. Using remote sensing products, physiological





modeling and crop data, Still et al (2003) found that $C_3$ and $C_4$ area abundances are 87.4 and 18.8 million $km^2$ (17.7 % as $C_4$) and $C_3$ and $C_4$ gross primary production were 114.7 and 35.3 $PgC\ yr^{-1}$ (23 % as $C_4$), respectively, on average for the 1980s and 1990s. More recently, Luo et al (2024) used global observations of plant photosynthetic pathways, satellite remote sensing, and photosynthetic optimality theory to estimate a similar level of $C_4$ area coverage (17.5 %), but a lower fraction of gross primary productivity (19.4 %), compared to Still et al (2003). Luo et al (2024) showed that $C_4$ vegetation coverage decreased

from 17.7 % to 17.1 % over 2001 to 2019 as natural $C_4$ grass cover declined in favor of $C_3$ vegetation, especially C3 trees in tropical grasslands and savannas. In comparison, across the TRENDY ensemble of dynamic global vegetation models, there were large ranges of 7-23 % of vegetation area and 2-40 % of productivity from $C_4$ vegetation (Luo et al. 2024), showing a need for better constraints and understanding of $C_3/C_4$ vegetation competition and change over time.

Here we investigate the contribution of vegetation diversity representation to uncertainty in carbon flux simulation in Earth

System Models. We evaluate the representation of $C_3$ and $C_4$ vegetation over 1850-2014 in 11 Earth System Models used in the 6[th] Coupled Model Intercomparison project (CMIP6). We assess the distribution, productivity and carbon content of $C_3$ and $C_4$ vegetation and compare with observation-based estimates where possible. We also explore the potential trend in global carbon isotope discrimination $\Delta^{13}C$ due to $C_3$ and $C_4$ vegetation changes simulated in the models.

## 2 Materials and Methods

### 2.1 CMIP6 Models Outputs and the LUH v2 Data

To analyze the CMIP6 models, we obtained the output for the following variables from the CMIP6-ESGF repository (https://esgf-index1.ceda.ac.uk/projects/cmip6-ceda/): fractions of $C_3$ and $C_4$ vegetation coverage (c3PftFrac and c4PftFrac), $C_3$ and $C_4$ crop fraction (cropFracC3 and cropFracC4), total crop fraction (cropFrac), gross primary production (gpp),

vegetation carbon content (cVeg), soil carbon content (cSoil), grid cell area (areacella), and percentage of each grid cell covered by land (sftlf). The necessary output was available from 11 models (Table 1). Brief descriptions of the implementation of vegetation abundance in each of the 11 models are provided in the Supplementary resources (Materials and Methods S1).

**Table 1. Summary of each model's definition of crop cover and our calculation of $C_3$ and $C_4$ crop cover. More detail is given in SM Materials and Methods.**

| Earth System Model | Land Model | $C_3$ Crop Fraction | $C_4$ Crop Fraction | Definition of crop cover |
|---|---|---|---|---|
| ACCESS-ESM1.5 | CABLE | cropFracC3 | Zero everywhere | LUH2 mapped onto PFTs |
| CanESM5 | CLASS-CTEM | cropFracC3 | cropFracC4 | LUH2 mapped onto PFTs |




| CESM2 | CLM5.0 | cropFrac * c3PftFrac | cropFrac * c4PftFrac | LUH2 mapped onto PFTs |
|---|---|---|---|---|
| CESM2-WACCM | CLM5.0 | cropFrac * c3PftFrac | cropFrac * c4PftFrac | LUH2 mapped onto PFTs |
| CMCC-CM2-SR5 | CLM4.5 | cropFracC3 | Zero everywhere | LUH2 mapped onto PFTs |
| CMCC-ESM2 | CLM4.5 | cropFracC3 | Zero everywhere | LUH2 mapped onto PFTs |
| CNRM-CM6-1 | ISBA-CTRIP | cropFrac * c3PftFrac | cropFrac * c4PftFrac | ECOCLIMAP-II fixed |
| CNRM-ESM2.1 | ISBA-CTRIP | cropFrac * c3PftFrac | cropFrac * c4PftFrac | LUH2 mapped onto PFTs |
| MPI-ESM-1-2-HAM | JSBACH | cropFracC3 | cropFracC4 | LUH2 mapped onto PFTs |
| MPI-ESM1-2-LR | JSBACH | cropFracC3 | cropFracC4 | LUH2 mapped onto PFTs |
| UKESM1 | JULES-TRIFFID | cropFracC3 | cropFracC4 | LUH2 mapped onto PFTs with $C_3/C_4$ competition from TRIFFID |

To specify crop cover, ten of the eleven CMIP6 models use the Land Use Harmonization version 2 (LUH2) dataset (Hurtt et al 2017, Table 1) so we also analyze the LUH2 data directly here. LUH2 estimates the fractional area and transition of use of
12 categories at an annual and 0.25° by 0.25° spatiotemporal resolution, starting from the year 850 (Hurtt et al., 2020). Five of these categories are $C_3$ annual crops, $C_3$ perennial crops, $C_3$ nitrogen fixers crops, $C_4$ annual crops and $C_4$ perennial crops. We group these categories into $C_3$ crops and $C_4$ crops, and their sum as total crops.

The UKESM1 model uses the LUH2 crop cover data with some modifications including simulation of competition between $C_3$ and $C_4$ vegetation using the dynamic vegetation model TRIFFID (Top-down Representation of Interactive Foliage and
Floral Including Dynamics, Sellar et al., 2019, Clark et al., 2011). Other models use the LUH2 data more directly, but differences can result from mapping the LUH2 data onto the model's PFTs, land cover and grid maps.

In CNRM-CM6.1, the crop cover is based on the ECOCLIMAP-II database (Voldoire et al., 2019) CNRM-CM6-1 adopts a vegetation cover map that is fixed to its present-day distribution, so that temporal changes in crop area are not included. The reasoning for using a fixed distribution is that the carbon cycle is not fully resolved in the model and there is uncertainty in
impacts of land use and land cover change on carbon flux (Faroux et al., 2013; Voldoire et al., 2019).

## 2.2 Global Land Vegetation Cover, Productivity, and Carbon Content

We analysed changes in the vegetation cover for all 11 CMIP6 models by calculating the gridded annual area fraction for $C_3$ and $C_4$ vegetation for each year 1850-2014, which we also separated into crops, natural and total vegetation. We also calculated
gridded annual gross primary production (GPP), vegetation and ecosystem carbon content for $C_3$ and $C_4$ vegetation in crops, natural and total vegetation in each model. We combined gridded estimates into global totals.





We calculated annual maps of area fraction of $C_3$ crops (cropFracC3) and $C_4$ crops (cropFracC4). Most models included the cropFracC3 and cropFracC4 variables that could be used directly (Table 1). Some models had to be treated differently, due to the output provided. ACCESS-ESM1.5 and the CMCC models did not provide cropFracC4 and their cropFracC3 was equal to

the total crop fraction (cropFrac), so we specified the fraction of $C_4$ crops to be zero everywhere. The CESM2 models did not provide cropFracC3 and cropFracC4 variables, but they did provide c3PftFrac or c4PftFrac (the fractions for total vegetation), so cropFracC3 and cropFracC4 were calculated by multiplying cropFrac by c3PftFrac or c4PftFrac. In CNRM-CM6-1 and CNRM-ESM2.1, the cropFracC3 and cropFracC4 variables were identical to the c3PftFrac and c4PftFrac variables, which appears to be an error in the output. So in CNRM-CM6-1 and CNRM-ESM2.1 cropFracC3 and cropFracC4 were calculated

by multiplying cropFrac by c3PftFrac or c4PftFrac.

To calculate the fractions of natural $C_3$ and $C_4$ vegetation, we subtracted cropFracC3 and cropFracC4 from c3PftFrac or c4PftFrac (the total $C_3$ and $C_4$ vegetation fractions). Therefore, for each grid cell we had the fractional area of $C_3$ and $C_4$ crops and of $C_3$ and $C_4$ natural vegetation.

The GPP from $C_3$ or $C_4$ vegetation was not provided so we estimated these by multiplying total GPP with the area fractions.

We applied the same procedure to estimate vegetation carbon content and total ecosystem carbon content by vegetation type. In Luo et al. (2024) it was found that the per unit area photosynthetic rate of $C_4$ grass was generally higher than for $C_3$ vegetation, so these calculations may overestimate the proportion assigned to $C_3$ vegetation.

**2.3 Global Stable Carbon Isotopic Discrimination**

To estimate how the simulated changes in $C_3$ and $C_4$ fractions of GPP influenced the global isotopic discrimination, we estimated discrimination assuming fixed values for discrimination by $C_3$ and $C_4$ vegetation. This neglects any environmental or physiological effects on discrimination to isolate the potential effect from changes in $C_3$ and $C_4$ fractions of GPP alone. The annual global isotopic discrimination was calculated by the following weighted average with GPP:

$\Delta = (GPP_{C3} \Delta_{C3} + GPP_{C4} \Delta_{C4})/( GPP_{C3} + GPP_{C4})$ (1)

Here, $GPP_{C3}$ and $GPP_{C4}$ are the integrated GPP for all $C_3$ and $C_4$ vegetation each year, and $\Delta_{C3}$ and $\Delta_{C4}$ are the fixed stable carbon isotope discrimination for $C_3$ and $C_4$ respectively. Click or tap here to enter text. We also calculated the global natural isotopic discrimination ($\Delta_{nat}$) using the natural $C_3$ and $C_4$ GPP for all the models using the same weighted average approach. To calculate the effect of crops on the global annual isotopic discrimination ($\Delta_{crop}$), we subtracted $\Delta_{nat}$ from $\Delta_{tot.}$

$\Delta_{C3}$ and $\Delta_{C4}$ were specified as the means of $C_3$ and $C_4$ plant leaf isotopic discrimination observations from the database of 3987 species published by Cornwell et al. (2018), which are 20.7 ‰ for $\Delta_{C3}$ and 6.3 ‰ for $\Delta_{C4}$ (Fig. 1). The two peaks in the histogram are assumed to correspond to the highest count for the species with either the $C_3$ and $C_4$ photosynthetic pathways. A breakpoint of 12.5 ‰ was chosen for $C_3$ and $C_4$ based on the midpoint between the two peaks and then the means were calculated from the two distributions.






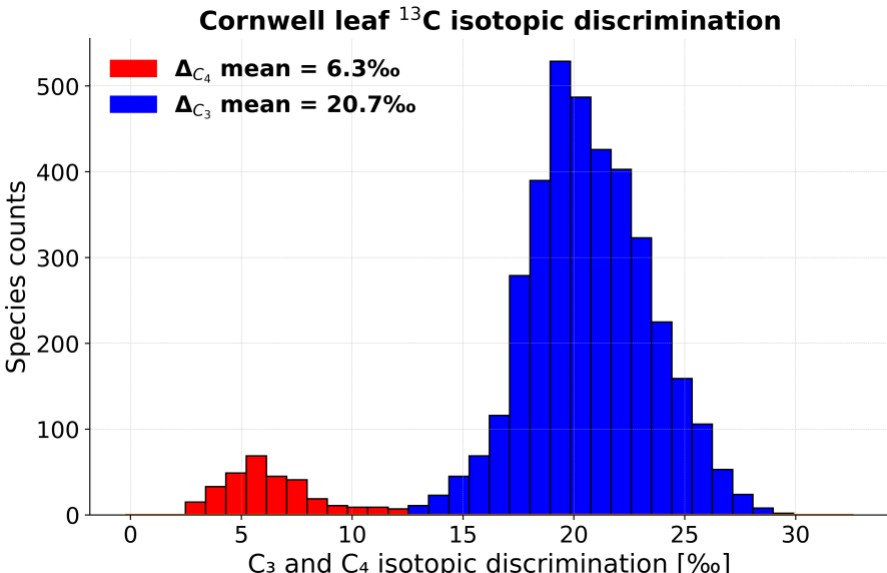

Figure 1: Stable carbon isotopic discrimination in leaves from database published by Cornwell et al (2018).

## 3 Result

### 3.1 Crop Abundance and its Change Over Time

Based on the LUH2 dataset averaged over 1970-2014, the highest $C_3$ crop abundance is in agricultural regions in central North America, southwest Europe, southeast and south Asia, Sub-Saharan Africa, southeast South America and southern Australia (Figures 2 and 3) (Hurtt et al., 2020). The fractional crop area is between 60 and 100 % in these regions. For $C_4$ crops, the highest abundance is in the Great Plains in North America, Sub-Saharan Africa, southwest Asia and southeast Asia (Fig. 2). The models that do not use LUH2 data directly, UKESM1 (which used LUH2 indirectly) and CNRM-CM6-1, show differences

with LUH2 (Figures 2 and 3). UKESM1 underestimates $C_3$ crops in Sub-Saharan and central Africa and in southeast Asia, particularly in India, which is likely due to a lack of precipitation in UKESM1 in these areas (Sellar et al 2019). UKESM1 estimates more $C_4$ crop coverage than LUH2, particularly in Asia (Figure 2 and 3 and Table S1). Crop coverage in UKESM1 is about 10 % lower than LUH2 between the equator and 30°N (Fig. 3). CNRM-CM6-1 has less $C_4$ crop cover in Europe and Africa, but more in North and South America, compared to LUH2 (Figures 2 and 3).

Between 1970 and 2014, there were decreases in crop coverage in North America, Europe, Southern Africa, Chile, Japan and New Zealand but increases in South and Southeast Asia, Africa and South America (Figure 3b and S1). The largest increases of 10 % or more were concentrated around the eastern region of south America, especially in Brazil, the southern fringes of the Sahara Desert (sub-Saharan Africa) and southeast Africa. The largest decreases of more than 20 % were across the corn-belt of the United States and in southwest and central Europe. Changes in crop area in UKESM1 were consistent with LUH2



except in the low latitudes of the Northern Hemisphere, where increases in C₃ crops in Africa and Asia were underestimated, compared to LUH2 (Figure 3b and S1).

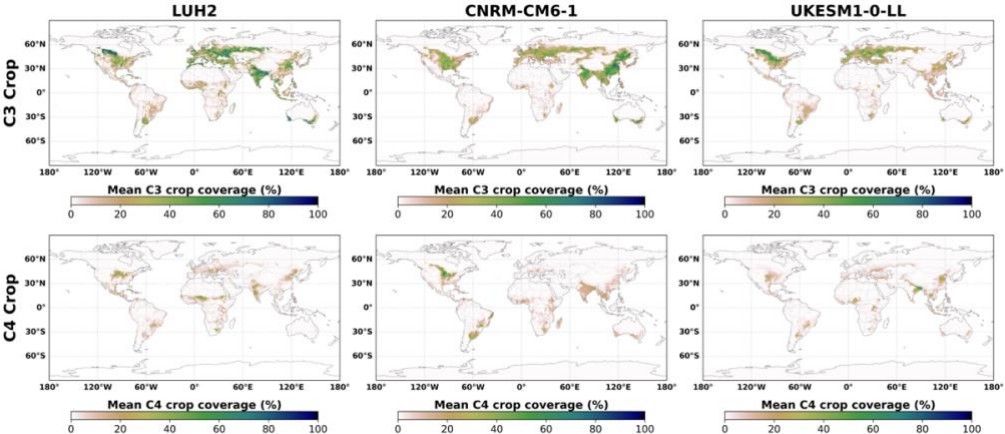

**Figure 2: Mean percentage of earth surface covered by Crops between 1970 and 2014 For C₃ and C₄ in LUH2 and UKESM1-0-LL.**
Other models not shown use the LUH2 crop variables directly for their crop coverage and change.

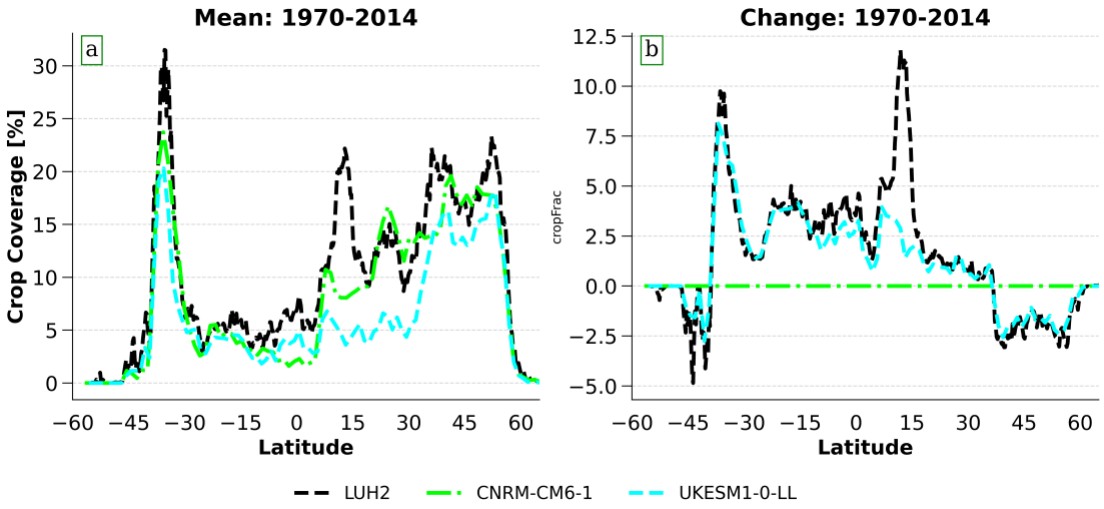

**Figure 3: Mean and change in crop coverage averaged over latitude between 1970 and 2014.** (a) Mean crop coverage (b) change in crop coverage.

**3.2 Global Trends in Vegetation Coverage**

The models incorporating LUH2 have similar patterns in C₃ and C₄ crop coverage over 1970-2014; however, the total area of crops in these models differed from LUH2 by -19 to +3 % (Fig. 4, Table S1). CanESM5 was the most consistent with LUH2. Inconsistences among the models that use the LUH2 dataset may be due to how the data were pre-processed before



incorporation into the model, due to grid spacing or due to differences in the model PFTs compared to LUH2 categories,
particularly as some models did not include $C_4$ crops. Also, since the variables cropFracC3 and cropFracC4 were not available for all models, our calculations of the crop fractions may have produced discrepancies. The magnitude of area abundance for $C_3$ and $C_4$ crop and natural vegetation for the years 1970 and 2014 are provided in the supplementary materials (Table S2).

There is a strong positive trend in the area of croplands from 1850 until 2014 (Fig. 4a). Total crop area rose by 200 % in LUH2 and models using LUH2 data. Crop area in UKESM1 was lower than LUH2 but increased in greater proportion, from 3 to 10
million $km^2$ (217 %). In CNRM-CM6 total crop area was fixed at ~14 million $km^2$, similar to LUH2 in the 1990s.

While representing a smaller fraction of total crop area (10-45%), the area of $C_4$ crops increased in greater proportion (>300 %) than $C_3$ crops (160 %) in the LUH2 data (Fig. 4). In the CESM2 models, $C_3$ crop area is lower and $C_4$ crop area is higher than in LUH2, despite the model using LUH2 data, although this may be affected by our calculation from the variables provided (Table 1). In UKESM1, $C_3$ crop area was much smaller than LUH2 while its $C_4$ crop area was larger (Fig. 4). UKESM1 $C_3$
crop area peaked between the 1980s and 90s before declining slightly in the 2000s, in contrast to LUH2 where it continued to increase after 2010. UKESM1 $C_4$ crop area rose through 2014, when it was about 42 % higher than LUH2. CNRM-CM6-1 had a fixed $C_4$ crop fraction of 24 %, slightly higher than the $C_4$ crop fraction in LUH2 in the 1990s. ACCESS-ESM1-5, CMCC-ESM2 and CMCC-CM2-SR5 do not have $C_4$ crops so their total crop area is equivalent to the $C_3$ crop area.

The area of natural vegetation decreases in all models from 1850 until 1940-70, when four models (CNRM-ESM2-1,
UKESM2, and the MPI models) start increasing while the others continue decreasing. Natural vegetation is similarly dominated by $C_3$, with 9-23 % of natural vegetation as $C_4$ in 2014, and most of the decline in natural vegetation area is in $C_3$ vegetation. The trend in natural $C_4$ vegetation area is inconsistent across the models. In CanESM5 and ACCESS-ESM1-5, the natural $C_4$ vegetation area decreased over 1850-2014, while for the MPI-ESM models the natural $C_4$ vegetation area increased especially from 1990. For the CESM2 and CMCC models, the natural $C_4$ vegetation area was constant until the early 2000s
before falling slightly. Compared to Luo et al (2024)'s estimate of the area of natural $C_4$ vegetation for 2000-14, UKESM1 is quite consistent, while other models simulate up to 15 % larger and up to 63 % smaller areas (Fig. 4h).





**Figure 4: Global temporal trends in vegetation area from 1850 to 2014 in CMIP6 models.** Trend of the area covered by (a) total crop, (b) total natural vegetation (c) total global vegetation, (d) $C_3$ crops, (e) $C_3$ natural vegetation, (f) total $C_3$ vegetation, (g) $C_4$ crop, (h) $C_4$ natural vegetation and (i) total $C_4$ vegetation. All models except CNRM-CM6-1 used LUH2 data to inform their crop coverage. ACCESS-ESM1-5, CMCC-ESM2 and CMCC-CM2-SR5 do not have $C_4$ crops.

For the total vegetation area, four models have distinct positive trends from the late 1960s through to 2014 (CNRM-ESM2.1, UKESM1 and the MPI models) (Fig. 4c), driven mostly by increase in $C_4$ natural vegetation (Fig. 4h). In the other models the increase in crop area is balanced by the decreased in area covered by natural vegetation. The estimated total vegetated area from the European Space Agency Climate Change Initiative (ESA-CCI) for 2000-2014 is matched by the CMCC models, while the CESM2 models simulate larger vegetated areas and all other models simulate smaller vegetated areas. The ESA CCI data shows a small increase that may be caused by the replacement of bare ground by natural grasses (Fig. S5).



### 3.3 Global Trends in Gross Primary Production

While all the models simulate an increase in GPP, the magnitude and contribution by vegetation type differ (Figure 5). The total GPP for all the models increased steadily until the 1960s and then grew sharply for the rest of the historical period (Fig. 5c). The increase in GPP in crops is linked to the area increase, while the GPP change in natural vegetation is decoupled from the change in area. In CNRM-CM6-1 with a fixed vegetation cover, the trend of $C_3$, $C_4$ and total crop GPP is not as strong compared to other models especially before the 1970 (Fig. 5a, d and g). They increased by 32 %, 27 %, and 18 % respectively

compared to the ensemble mean increase of 195 %), 251 % and 209 % respectively (Fig. 5a, d and g). The increasing GPP trend in natural vegetation is likely dominated by the $CO_2$ fertilization effect in $C_3$ vegetation. The models disagree on the magnitude of total global GPP, ranging from 80 PgC yr$^{-1}$ in CNRM- ESM2.1 to 150 PgC yr$^{-1}$ in MPI-ESM (-20 to + 29 %) in 2014  (Gier et al., 2024; Arora et al., 2020) (Fig 6c).

        For GPP in natural vegetation, there are large ranges of 54-100 PgC/yr simulated for $C_3$ vegetation and 9-23 PgC/yr simulated

for $C_4$ vegetation across the models before 1970 (Fig. 5e and h). Despite the decrease in natural $C_3$ vegetation area before 1970, the $C_3$ GPP trend is generally either weakly negative or unchanging until the 1970 (Fig. 5e). Then all models increase steadily over 1970-2014. In natural $C_4$ vegetation, there is generally an increase in GPP since 1850, but the increase is not at the same rate in all the models. It is weaker in ACCESS-ESM1-5, CanESM5 and UKESM1 (Fig. 5h). Three models show a decrease in natural $C_4$ GPP over 2000-14 (ACCESS-ESM1-5 and the CESM2 models).

The proportion of total GPP by $C_4$ vegetation in the models is 12-27 % (Fig. 5i and S7), compared to 23 % in Still et al. (2003) and 19 % in Luo et al (2024). Compared to other model simulations, the range was 2-40 % in TRENDY models (Luo et al. 2024), and 18-27 % in previous modelling studies (Farquhar & Lloyd, 1993; Fung et al., 1997).





**Figure 5: Temporal trend in gross primary production (GPP) between 1850 and 2014.** Trend of GPP of (a) total crop, (b) total natural vegetation (c) total global vegetation, (d) $C_3$ crops, (e) $C_3$ natural vegetation, (f) total $C_3$ vegetation, (g) $C_4$ crop, (h) $C_4$ natural vegetation and (i) total $C_4$ vegetation.

## 3.4 Trends in Global Vegetation and Land Carbon Stock

The vegetation carbon stock ($C_{veg}$) in crops increased consistently in 10 of the 11 models, at least doubling in magnitude for the historical period. This increase was generally less than the increase in area (Fig. 4a) and GPP (Fig. 5a). In CNRN-CM6-1 that used fixed vegetation cover, the $C_{veg}$ in crops decreased until the late 1990s before increasing slightly for the rest of the period. The magnitude of vegetation carbon content in crops ranged widely, from 26 PgC in CMCC-ESM2 to 66 PgC to in CNRM-CM6-1 in 2014 (Fig. 6a).



In natural vegetation, most models simulated a negative trend in $C_{veg}$ until the late 1970s before reversing the trend and
increasing sharply for the remainder of the historical period (Fig. 8b). In two models, natural $C_{veg}$ was relatively constant until
the 1970s (CNRM-CM6-1 and CMCC-ESM2). Again, the models do not agree on the magnitude of natural $C_{veg}$, which ranges
from 328 PgC in MPI-ESM1.2-LR to 573 PgC in ACCESS-ESM1.5 with a mean of 415.04 PgC in 2014 (Fig. 6b).

**Figure 6: Temporal Trend in global vegetation carbon content between 1850 and 2014.** Absolute values of carbon content in (a) crops,
(b) natural vegetation and (c) total vegetation. Carbon accumulation since 1850 in (d) crop, (e) natural vegetation and (f) total vegetation.
Gibbs and Ruesch (2008) above and below ground vegetation carbon stock estimate of 492.3 PgC is based on a map created using the
International Panel on Climate Change (IPCC) Good Practice Guidance for reporting national greenhouse gas inventories, to serve as a
benchmark to guide policies aimed at reducing carbon emissions from land-use change and also enhance simulation of carbon stock in ESMs.
Spawn et al (2020) estimate the above and below ground global terrestrial vegetation biomass at 423.3 PgC by merging different landcover-
specific global maps generated from remote sensing products into one global vegetation carbon stock map at 300m spatial resolution for the
year 2010. Erb et al (2018) estimated that the actual mean global carbon stock is 450 (range of 380 to 536 PgC) by analysing the impact of
vegetated land-cover conversion and land management on the global carbon state of the biosphere ecosystems.





### 3.5 Global Trends in Stable Carbon Isotope Discrimination

The models disagree on the magnitude and trend of $\Delta_{tot}$, $\Delta_{nat}$ and $\Delta_{crop}$ (Fig 7), based on fraction of GPP from C$_3$ and C$_4$ vegetation (Equation 1). As a result of the 12-27 % range in C$_4$ fraction of total GPP, the global total discrimination ranges from 16.9 to 19.5 ‰ (Table S2) in 2014 across the models, compared to 16.5 ‰ in Still et al. (2003). Models simulate no change (CESM2 models), slightly increasing $\Delta_{tot}$ (ACCESS-ESM1.5 and CNRM-CM6.1) or decreasing $\Delta_{tot}$ (all other models). The strongest decreases in $\Delta_{tot}$ were in the MPI models which had the strongest increases in C$_4$ vegetation area (Fig. 7).

Compared to the strong positive trend in global discrimination trend derived by Keeling et al (2017) from atmospheric $\delta^{13}$C data, the effect of vegetation change in the models was much smaller or had the opposite sign (Fig 7).

The models also disagree on the sign and magnitude of the trends in $\Delta_{nat}$ (17 to 19.42 ‰) and $\Delta_{crop}$ (-0.7 to 0.33 ‰, Table S2). Among models with decreasing $\Delta_{tot}$, the trend is primarily driven by $\Delta_{nat}$ in the MPI models but by $\Delta_{crop}$ in UKESM1 and CanESM5 models.


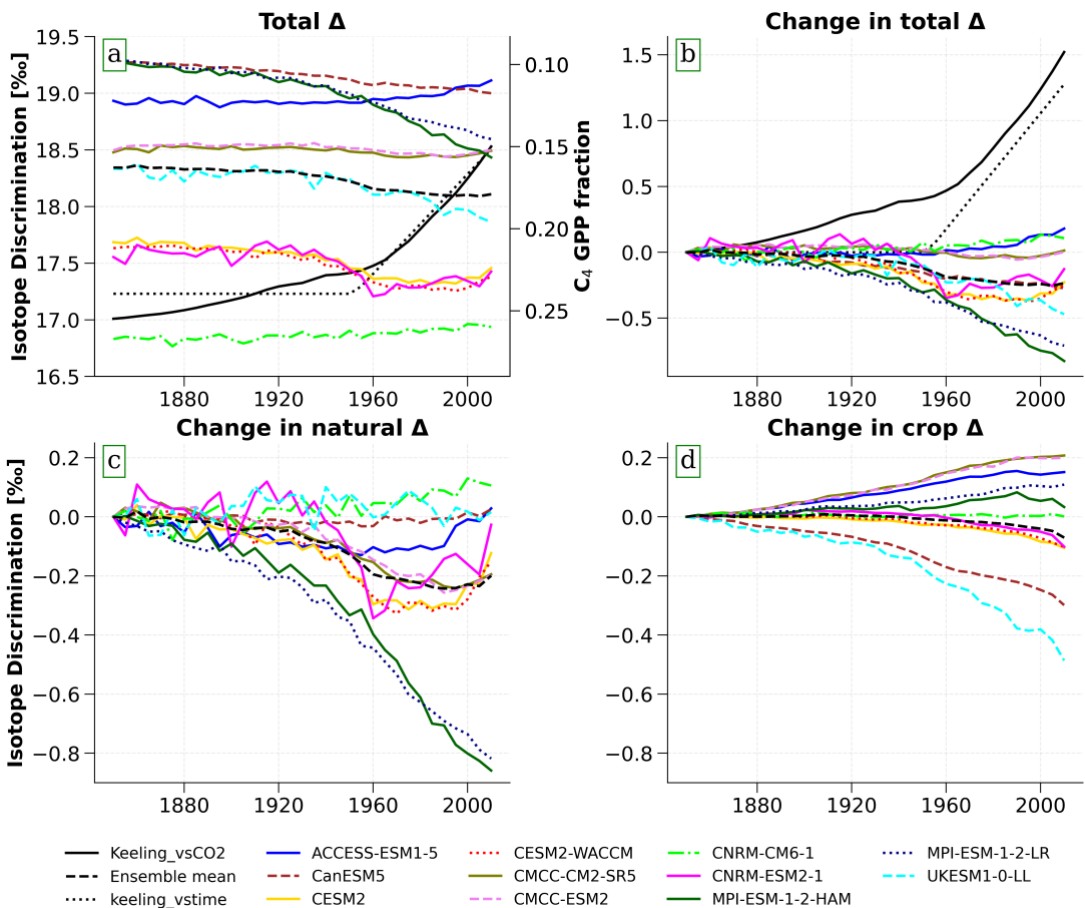

**Figure 7: Trend of global stable carbon isotopic discrimination between 1850 and 2014.** 10-year average discrimination of (a) total vegetation ($\Delta_{tot}$), (b) difference in $\Delta_{tot}$ from 1850, (c) component of difference due to natural vegetation ($\Delta_{nat}$) and (d) due to crops ($\Delta_{crop}$).



## 4 Discussion

By examining the components of global vegetation area, GPP and carbon stocks by vegetation type, we find that CMIP6 models span a large range for nearly all variables. Some models showed stronger gains in GPP and carbon stocks in natural $C_3$ vegetation, while for others the strongest gains were in either crop or natural $C_4$ vegetation. Therefore, better quantification of vegetation types and more consistency in models would improve future carbon modelling. For example, total vegetated areas in some models that are much lower than the ESA-CCI estimate (Harper et al., 2023) and total $C_4$ vegetated areas in some

models that are much lower than the Luo et al. (2024) estimate can probably be ruled out. However, the spread across models in many cases reflects the lack of observational constraints.

Even though nearly all of the models use LUH2 for input data, their crop area and particularly its attribution to $C_3$ and $C_4$ vegetation are not consistent. While most models underestimated total crop area compared to LUH2 (Fig. 4a), another data-based cropland map (Potapov et al., 2022) estimates lower crop areas that overlap some of the CMIP6 models' crop areas.

Therefore, the spread in crop area in the models may be consistent with the uncertainty in actual crop area. A limitation of some current models is that they do not include $C_4$ crops, this carbon fluxes associated with $C_4$ crops cannot be simulated in those models. In future, the total crop area is likely to increase (Molotoks et al., 2018; O'Neill et al., 2016) so that agriculture will become an even stronger influence on carbon fluxes.

While increasing crop area leads to increased GPP in crops, this does not translate to an increase in total carbon stocks in

vegetation because the expansion of croplands leads to reduction of forested land and $CO_2$ emissions (Gasser et al., 2020; Lam et al., 2021; Luo et al., 2024). West et al (2010) estimated that the replacement of a unit of natural vegetated land by crop will lead to the loss of double the amount of $CO_2$ uptake in croplands (-12 million kg km$^{-2}$ vs. +6.3 million kg km$^{-2}$). As such, in some models the accumulation of carbon in crop vegetation did not outweigh the loss in natural vegetation over 1850-2014 and there was a net loss of carbon. In other models there was a net loss followed by a net gain in recent decades, due to strong

gains in carbon in natural vegetation (Fig. 7). Observation-based estimates also disagree on the carbon stock changes in vegetation, with Bar-On et al. (2025) finding little to no change in vegetation carbon stocks over 1992-2019 while (Pan et al., 2011; 2024) found significant gains.

The range in global GPP was even larger than the range in vegetated area across the models, both for total (GPP: -23 to +37 % vs area: -11 to +8 %, compared to the model mean) and for $C_3$ (GPP: -24 to +32 vs area: -19 to +8 %) vegetation types. In

$C_4$ natural vegetation, there was also disagreement over the trend in GPP, with some models showing increases in GPP in grid cells dominated by $C_4$ (>75 %, Fig. S6; a-d), and others showing decreases. These inconsistencies indicate that models' parametrizations, simulated climate and other factors (Campbell et al., 2017; Hou et al., 2022; Lavergne et al., 2022; Zscheischler et al., 2014) are at least as important as vegetation area for GPP simulation. Therefore, achieving consistency in area for vegetation types is not sufficient and improved understanding and representation of $C_3$ and $C_4$ GPP in models is

needed.





The range in total area fraction of $C_4$ vegetation of 10-26 % in the CMIP6 models spans the observation-based estimates of 17.1-17.7 % (Still et al. 2003; Luo et al. 2024) and has a similar range as the dynamic global vegetation models in the TRENDY project (7-23 %; Luo et al. 2024). Since the observation-based estimates are quite consistent, model simulations of carbon fluxes can likely be improved by detailed comparison and specification of $C_4$ vegetation cover in particular PFTs in models to

improve their correspondence to observational estimates.

The trends in $C_3$ vs $C_4$ contributions to GPP simulated by CMIP6 models produced differing trends in isotopic discrimination (Fig. 7). They did not produce strong increasing trends as found by analysis of atmospheric $\delta^{13}C$ data (Keeling et al. 2017; Peters et al. 2018). Since the effect of vegetation change on discrimination varied widely across the CMIP6 models, it is still uncertain how much vegetation changes could add to or oppose changes in discrimination caused by environmental or

physiological effects. In particular, since the GPP trend in the models (11 to 18 %, Fig. S8) is weaker than the $CO_2$ fertilization effect over the 20[th] century based on carbonyl sulphide data (+30 %; Campbell et al. 2017), there may have been a more positive $C_3$ vegetation-driven trend in discrimination in reality than in the models. Since the CMIP6 models did not provide output for GPP and carbon stocks for $C_3$ and $C_4$ vegetation or for crops and natural vegetation, we had to scale total GPP and carbon stocks by the area fractions of each vegetation type for grid cells with mixed vegetation. This calculation may not

represent the models' actual attribution precisely, since $C_3$ vegetation typically has higher carbon density and $C_4$ vegetation can have higher GPP relative to area (Luo et al. 2024). Incorporating more detailed vegetation information from the models would improve the accuracy of the calculations.

## 6 Discussion

We analysed vegetation changes for $C_3$ and $C_4$ photosynthetic pathways in natural vegetation and crops in 11 CMIP6 models

over the historical period of 1850 to 2014. Except for one model that used a fixed vegetation distribution, the models include the expansion of agriculture using LUH2 data. Still, there is a significant variation in the fraction of area and GPP allocated to crops in these models, and the UKESM1 model strongly underestimated crops in Asia and Africa, likely due to biases in simulated climate. The $C_4$ fraction of vegetated area has remained relatively constant in the models, though there is a large range in simulated $C_4$ area fraction of 9 to 13 million $km^2$ in 2014 while observation based estimates are quite consistent at

17.1 - 17.7 % (Luo et al. 2024; Still et al. 2003). Overall, we found that the total vegetation area in most models is not changing, but the global vegetation composition is changing in favour of crops. Expansion of crops has negative implications for the carbon sink capacity of the terrestrial biosphere. Whereas all the models agree that $C_3$ vegetation and total global GPP is increasing, the magnitude of the increase spans a large range. In $C_4$ vegetation, the models disagree magnitude of GPP but also the sign of its trend, especially in natural vegetation, implying that the model parametrizations for the simulation of $C_4$ GPP

need improvement. The strong positive trend in UKESM1 $C_4$ crop and MPIs $C_4$ natural vegetation area clearly drives the decline of their global discrimination trend, an effect which is not as pronounced in other models' vegetation categories, however, due to the large uncertainty in vegetation area abundance and GPP, it will be difficult to accurately determine how

consistent this influence is on global isotopic discrimination trend. The mean of vegetation carbon contents in the model generally agrees with observation-based estimates. We have shown in this study that carbon flux simulation in current ESMs

includes uncertainties in different components, reducing these uncertainties will require more observational constraints and more robust and realistic model representations.

**Appendices**

Appendix A contains the supplementary material.

**Code Availability**

The code (Jupyter notebooks) used for the analysis of the CMIP6 models data and for the data visualization can be found in the following Github repository: https://github.com/jovwemuvwose/CMIP6_Model_Analysis_Project_2025 and Zenodo at: https://zenodo.org/records/16883407

**Data Availability**

The CMIP6 models data used in the are available at the World Climate Research Programme (WCRP) Coupled Model

Intercomparison Project (Phase 6) website: https://esgf-node.ipsl.upmc.fr/projects/cmip6-ipsl/. The leaf carbon isotope data by Cornwell et al (2018) are available at https://onlinelibrary.wiley.com/doi/10.1111/geb.12764. For the ESA CCI, LUH2, Still, and Luo data used in Figure 4, see Harper et al (2023), Hurtt et al (2020), Still et al (2003) and Luo et al (2024) respectively. For Gibbs, Spawn and Erb data used in the Figure 6 above see Gibbs and Ruesch (2008), Spawn et al (2020), Erb et al (2018) respectively and Graven et al (2024) for all these three data in in one file at

https://www.science.org/doi/10.1126/science.adl4443. For the Keeling_vsCO2, Keeling_vstime used in Figure 7 see Keeling et al 2017. And for the GCB data used in Figure S4 see Friedlingstein et al (2025). The processed data used for the figures can be made available by the authors on request.

**Authors contribution**

JO and HG conceptualize the overarching research and CIP contributed to broadening it; JO prepared and created the published

work, specifically data analysis and visualization and writing of the initial draft of the manuscript. HG and CIP reviewed and edited the manuscript



**Competing interests**

The authors declare that they have no competing interest.

**Acknowledgement**

This project is supported by Schmidt Sciences, LLC.

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
