# Peer review of "Uncertainty in Land Carbon Fluxes Simulated by CMIP6 Models from Treatments of Crop Distributions and Photosynthetic Pathways"

_EGUsphere, 2025_

## Referee Comment (RC1)

The paper studied the impact of varying representation of C3/C4 fractions across models on their photosynthesis and vegetation carbon simulations. They analyzed output from 11 CMIP6 models and found large spread in C3/C4 area, GPP and Cveg across models. However, the study relied on some assumptions that largely introduced bias, especially regarding the C3/C4 crop fractions and GPP calculations. I recommend a major revision and suggest authors to reconsider how to deal with those models without these outputs provided.

**Major Comments:**

1. For models lacking explicit C3/C4 crop fractions, authors used cropFrac × c3PftFrac or cropFrac × c4PftFrac to obtain the values. This can propagate large biases into estimates of C3/C4 crop area. For example, in a grid with 90% C3 tree and 10% C4 crop, you'll get C3 crop = 0.09 and C4 crop = 0.01, which is totally wrong. Although the authors already acknowledged this issue, this process was problematic and largely influenced the reliability of the findings.

2. Also, attributing C3/C4 GPP using the product of total GPP with C3/C4 fractions is problematic. Although the authors noted this overestimated C3 GPP, the issue was broader and could influence all subsequent analyses, e.g., range of C4 GPP (12–27% in Fig. 5i).

3. Section 3.3 states "The increase in GPP in crops is linked to the area increase, while the GPP change in natural vegetation is decoupled from the change in area". This is an interesting point, but I did not see any evidence supporting this statement. I also suggest authors to reframe their study to discuss why models all dependent on LUH2 still disagree strongly on area and fluxes.

**Minor Comments**

Line 13: Change "affect" to "affects"

Line 18: What does this mean by 'the data-based estimate'?

Line 19: I did not see the value of "17%" over the Results

Line 85: Clarify how cSoil was used

Table 1: Provide the spatial resolution of each model

Line 101: It would be helpful to provide details about the preprocessing differences among models that all relied on LUH2

Line 103: Ensure consistent use of CNRM-CM6-1 or CNRM-CM6.1.

Line 110: This paper did not include any analysis about 'ecosystem carbon content'

Line 143: Why was 12.5 ‰ chosen for isotopic discrimination? Is this value spatially robust?

Line 150: Why is the isotopic analysis based on 1970–2014, while other analyses span 1850–2014?

Line 156: What does this mean by 'lack of precipitation in UKESM1'

Figure 2: Change "earth" to "Earth" and "For" to "for." Also, subfigures should be labeled (a)–(e).

Figure 3: Clarify the y-axis label

Line 177: The number (−19 to +3%) differs from the Abstract

Line 194–195: Provide full names of the MPI models. Also, use UKESM1 rather than UKESM2.

Figure 4: Provide data sources for ESA CCI, Still, and Luo.

Line 223: Should be Fig. 5c, not Fig. 6c and Line 245: Should be "Fig. 6b."

Line 285: how uncertainty in actual crop area is quantified?

Line 298–305: This could also be related to differences in the natural vegetation composition

---

## Author Comment (AC1)

**Uncertainty in Land Carbon Fluxes Simulated by CMIP6 Models from Treatments of Crop Distributions and Photosynthetic Pathways.**

Joseph Ovwemuvwose1, I. Colin Prentice2, 3, and Heather. D. Graven1

Correspondence to: Joseph Ovwemuvwose (j.ovwemuvwose22@imperial.ac.uk & josovw@gmail.com)

We are grateful to the referees for their comments and suggestions, which we have used to improve the manuscript. Below we describe the modifications we have made to the manuscript in response to the comments.

**Responses to Referees**

**Referee 1 Major Comments and Responses**

**Comment 1**

For models lacking explicit C3/C4 crop fractions, authors used cropFrac  $\times$  c3PftFrac or cropFrac  $\times$  c4PftFrac to obtain the values. This can propagate large biases into estimates of C3/C4 crop area. For example, in a grid with 90% C3 tree and 10% C4 crop, you'll get C3 crop = 0.09 and C4 crop = 0.01, which is totally wrong. Although the authors already acknowledged this issue, this process was problematic and largely influenced the reliability of the findings.

**Response to Comment 1**

This is a good point. To evaluate biases in the estimated values of  $C_3$  and  $C_4$  crops obtained from cropFrac  $\times$  c3PftFrac and cropFrac  $\times$  c4PftFrac in the models where this was applied (CESM2, CESM2-WACCM, CNRM-ESM2-1 and CNRM-CM6.1) we conducted a reliability analysis. We determined the similarity between the calculated distribution of  $C_3$  and  $C_4$  crops and LUH2 distribution using kernel density estimates (KDE) (Silverman 1986, Chen 2017) and spatial probability distributions (Chiang et al., 2021). We also visually compared the spatial distribution of these values to the LUH2, UKESM1 and MPI-ESM-1-2-HAM - two models that provided their  $C_3$  and  $C_4$  crop fractions to check if they are consistent. Our results show that the values obtained from the calculation are not inconsistent with other models and LUH2 in terms of magnitude, range and spatial distribution. The results are added to the supplement in Figures S11-S17. We added a description of these results to the main text in section 2.2.

Given the variables available in the CMIP archive, we had to make the calculation cropFrac  $\times$  c3PftFrac or cropFrac  $\times$  c4PftFrac to enable us to do a comprehensive model intercomparison. We have added a note to the discussion that models' reporting of their  $C_3$  and  $C_4$  crop fractions (cropFracC3 and cropFracC4) would allow for more robust analysis of the role of croplands in carbon flux simulation and the differences in the photosynthetic pathways.

**Comment 2**

Also, attributing C3/C4 GPP using the product of total GPP with C3/C4 fractions is problematic. Although the authors noted this overestimated C3 GPP, the issue was broader and could influence all subsequent analyses, e.g., range of C4 GPP (12–27% in Fig. 5i).

**Response to Comment 2**

We agree with the referee that the analysis would be improved with specific information on C3/C4 GPP from the models. However, we could only make use of the variables made available by the modelling groups, and this information was unfortunately not available.

<sup>1Department of Physics, Imperial College London, London, UK.

<sup>2Georgina Mace Centre for the Living Planet, Department of Life Sciences, Imperial College London, London, UK

<sup>3Department of Earth System Science, Tsinghua University, Beijing, China

We noted that the calculations may overestimate the proportion of GPP assigned to C3 vegetation, based on the findings of Luo et al. 2024. However, in the absence of full information, our approach should provide a reasonable approximation – because in pixels where plants with both pathways co-occur, the differences in GPP between C3 and C4 components will generally not be large due to the extra cost invested by C4 vegetation in the CO2 concentrating process (Still et al, 2000, Ehleringer and Bjorkman 1976, and Ehleringer 1978). Nonetheless, it is unfortunate that the modelling groups did not provide enough information for us to do a more exact analysis. We suggest that the GPP and Cveg of C3 and C4 vegetation should be explicitly reported in future. This relationship will likely become even less straightforward when scaled beyond the canopy level.

**Comment 3**

Section 3.3 states "The increase in GPP in crops is linked to the area increase, while the GPP change in natural vegetation is decoupled from the change in area". This is an interesting point, but I did not see any evidence supporting this statement.

**Response to Comment 3**

We have revised the manuscript to make the evidence for this point clearer.

If you look at Figure 4 (a) for instance, total crop area increased by 200 % compared to Figure 4 (b), in which natural vegetation on average decreased by 7 %. Then compare the changes in the vegetation area in the above figures to the change in the GPP in Figure 5 (a) and (b). Total crop GPP increased by 209% on average, however, for CNRM-CM6.1 with a temporally fixed vegetation cover, the increase in GPP is only 18 %. For natural vegetation, even though there is a mean 7 % decrease in total natural vegetation cover, we see on average  $\sim$ 18 % increase in GPP. This increase is likely mostly driven by the CO2 fertilization effect on vegetation with the C3 photosynthetic pathway.

We have created a figure (Figure S9 - S10) showing regression lines and corelation coefficients demonstrating the relationship between the vegetation area and GPP for crop and natural vegetation.

I also suggest authors to reframe their study to discuss why models all dependent on LUH2 still disagree strongly on area and fluxes.

We have added some more text regarding the disagreement with LUH for models incorporating LUH data in Lines 175 - 182. We also added more information in the supplementary material.

**Referee 1 Minor Comments and Responses**

Line 13: Change "affect" to "affects"

Done

Line 18: What does this mean by 'the data-based estimate'?

This has been changed to satellite-based estimate from Potapov et al 2022.

Line 19: I did not see the value of "17%" over the Results

This is based on Luo et al 2024 see Figure 4 (i) and Luo in the reference and the correct value is compared to 20  $\pm$  3 %.

Line 85: Clarify how cSoil was used

Soil carbon content is calculated for C3 and C4 cropland and natural vegetation. The results are available in the supplementary material. We have made it clearer how cSoil was used and what is in the supplementary material.

Table 1: Provide the spatial resolution of each model

We have added the spatial resolution of each model.

Line 101: It would be helpful to provide details about the preprocessing differences among models that all relied on LUH2

Some details about this are available in the supplementary material, but we are unable to provide information on all pre-processing that may have been done for each model.

Line 103: Ensure consistent use of CNRM-CM6-1 or CNRM-CM6.1.

**Corrected.**

**Line 110: This paper did not include any analysis about 'ecosystem carbon content'**

Response: This is referring to carbon content in vegetation and soil. It has been restated as 'vegetation carbon content and land carbon content' for clarification in the final manuscripta and the figure showing the result for this analysis is in Figure S3 and Figure S4.

**Line 143: Why was 12.5 ‰ chosen for isotopic discrimination? Is this value spatially robust?**

Response: 12.5 ‰ is the midpoint between the two peaks in Figure 1 from the database of leaf carbon isotope discrimination published by Cornwell et al (2018). The values to the left of 12.5 ‰ in the figure correspond to the values of stable carbon isotope discrimination in vegetation with the  $C_4$  photosynthetic pathway while values to its right correspond to stable carbon isotope discrimination in vegetation with the  $C_3$  photosynthetic pathways The mean values of these are then calculated and used for the analysis. They are shown in Figure 1: 6.3 ‰ for  $C_4$  and 20.7 ‰ for  $C_3$ . We have revised the text to clarify this point.

**Line 150: Why is the isotopic analysis based on 1970–2014, while other analyses span 1850–2014?**

Response: The isotope analysis spans the whole of the period from 1850 to 2014, as shown in Figure 7. We used 1970–2014 for C4, C3 and total crop in this section (Line 155 to Line 183) to focus on the period of largest change since the Green Revolution and how that change is captured in the CMIP6 models.

**Line 156: What does this mean by 'lack of precipitation in UKESM1'**

Response: In the Sellar 2019 reference cited, it is shown that the UKESM1 underestimates precipitation in India. We have revised the text to make this clearer.

Figure 2: Change "earth" to "Earth" and "For" to "for." Also, subfigures should be labelled (a)–(e). Done.

Figure 3: Clarify the y-axis label Done.

**Line 177: The number (-19 to +3%) differs from the Abstract**

This is good point. The values of -28 to +10 % from the satellite that is quoted in the abstract was not in the body of the text and so could mislead the reader. However, the -19 to +3 % mentioned here is a different value comparing models crop area to LUH2 crop area, which is different from the satellite-based estimate. We have now included both in the text to avoid confusion, as follows: "however, the total area of crops in these models differed from LUH2 by -19 to +3 % and it also ranged from -28 to +10 % of satellite-based estimate (Fig. 4, Table S1)"

Line 194–195: Provide full names of the MPI models. Also, use UKESM1 rather than UKESM2. Done

**Figure 4: Provide data sources for ESA CCI, Still, and Luo.**

The data sources are provided in the Data Availability section with the following statement: 'For the ESA CCI, LUH2, Still, Potapov and Luo data used in Figure 4, see Harper et al (2023), Hurtt et al (2020), Still et al (2003), Potapov et al (2022) and Luo et al (2024) respectively. For Gibbs, Spawn and Erb data used in the Figure 6 above see Gibbs and Ruesch (2008), Spawn et al (2020), Erb et al (2018) respectively and Graven et al (2024) for all these three data in in one file at https://www.science.org/doi/10.1126/science.adl4443. For the Keeling\_vsCO2, Keeling\_vstime used in Figure 7 see Keeling et al 2017. And for the GCB data used in Figure S4 see Friedlingstein et al (2025)'. The data can be accessed through the papers in the references.

Line 223: Should be Fig. 5c, not Fig. 6c Corrected.

and Line 245: Should be "Fig. 6b."

Line 273: Corrected.

Line 285: how uncertainty in actual crop area is quantified?

Response: This has been rephrased as 'the uncertainty in input crop area data source.'

Line 298–305: This could also be related to differences in the natural vegetation composition

This point has been added to the manuscript 'and may be also be linked to iscrepancies in the natural vegetation composition'.

**References**

The references for all the works cited here have been added to the main manuscript.

---

## Author Comment (AC2)

**Uncertainty in Land Carbon Fluxes Simulated by CMIP6 Models from Treatments of Crop Distributions and Photosynthetic Pathways.**

Joseph Ovwemuvwose1, I. Colin Prentice2, 3, and Heather. D. Graven1

Correspondence to: Joseph Ovwemuvwose (j.ovwemuvwose22@imperial.ac.uk & josovw@gmail.com)

We are grateful to the referees for their comments and suggestions, which we have used to improve the manuscript. Below we describe the modifications we have made to the manuscript in response to the comments.

**Responses to Referees**

**Referee Major Comments and Responses**

**Comment 1**

Core assumptions about C3/C4 GPP and Cveg. CMIP6 models do not directly report GPP or Cveg partitioned into C3 and C4 components. The authors therefore estimated C3 and C4 values by multiplying total GPP or Cveg by the areal fraction of C3 or C4 vegetation. As they themselves noted, this is problematic: on average, GPP per unit area is higher for C4 than for C3 vegetation, whereas Cveg per unit area is likely greater for C3 plants (often woody) than for predominantly herbaceous C4 plants. These relationships also vary geographically, shaped by climate and soil. Consequently, it is difficult to place much confidence in downstream analyses of C3/C4 GPP, Cveg, and their trends.

**Response**

We accept this point. We state that 'In Luo et al. (2024) it was found that the per unit area photosynthetic rate of  $C_4$  grass was generally higher than for  $C_3$  vegetation, so these calculations may overestimate the proportion assigned to  $C_3$  vegetation.' However, we can only make use of the variables that were made available by the modelling groups. We have applied a particular analytical approach in a consistent manner to all the models, so our findings do indeed reveal inconsistencies among the models.

Moreover, in the absence of full information, our approach should provide a reasonable approximation – because in pixels where plants with both pathways co-occur, the differences in GPP between  $C_3$  and  $C_4$  components will generally not be large due to the extra cost invested by  $C_4$  vegetation in the  $CO_2$  concentrating process (Still et al, 2000, Ehleringer and Bjorkman 1976, and Ehleringer 1978). In forested and other locations dominated by woody  $C_3$  vegetations, the biomass per unit area will be larger for  $C_3$  but in locations that are mosaics of both  $C_3$  and  $C_4$  grasses, there will be a small difference in the annual relative biomass of  $C_3$  and  $C_4$  since the relative distribution of their biomass is controlled by the seasonal availability of water (Winslow et al 2003). Nonetheless, it is unfortunate that the modelling groups did not provide enough information for us to do a more exact analysis. We suggest that the GPP and Cveg of  $C_3$  and  $C_4$  vegetation should be explicitly reported in future.

We agree that  $C_3/C_4$  competition varies geographically and is shaped by climate and soil properties. However, the objection raised here applies only to the approximate calculations we made within grid cells. We can reasonably expect the models to incorporate environmental and land-use controls on this competition appropriately, in order to represent geographic variations among pixels. The large inconsistencies we see suggest that models need to address this issue more rigorously.

A similar concern applies to the treatment of C4 crops. Several CMIP6 models lack explicit fractions for C4 crops versus natural C4 vegetation, so the authors assume C4 crop fraction = C4 fraction × total crop fraction. This implies that C3/C4 ratios for crops mirror those of natural vegetation at each grid cell—an unlikely scenario. Open data (e.g., Luo et al., 2024) show no such pixel-level pattern.

**Response**

<sup>1Department of Physics, Imperial College London, London, UK.

<sup>2Georgina Mace Centre for the Living Planet, Department of Life Sciences, Imperial College London, London, UK

<sup>3Department of Earth System Science, Tsinghua University, Beijing, China

Please see our response to Referee 1, Comment 2, on this issue. We have compared our approach to LUH and found that they are consistent.

The LUH C4-crop fraction itself is also problematic: in LUH c4 crop fraction remains static (based on ~2000 data), so the apparent change in C4 crop area likely reflects total cropland expansion or contraction. It is unclear how individual CMIP models treat this issue, and I would not regard the LUH C4 crop trend as an "observation," nor use it to declare a model (e.g., UKESM1) incorrect (L155, L328).

**Response**

Thank you for this comment. We have added a statement about the LUH C4 crop fraction remaining static, and revised the text so as *not* to imply that LUH is an observation that could invalidate a model.

If the aim is a robust model–data comparison of C3/C4 fractions in area, GPP, and Cveg, I suggest considering the TRENDY DGVM ensemble, which may provide explicit estimates for these variables.

**Response**

We consulted the TRENDY output available from trendyv12-gcb2023 at <a href="https://mdosullivan.github.io/GCB/">https://mdosullivan.github.io/GCB/</a>. Indeed, several of the models (CLM5.0, JULES, LPJ-GUESS, LPJ-ML, OCN, SDGVM, VISIT, YIBS, LPX-Bern and JSBACH) have some PFT-specific variables available but these are not well defined for both C3 and C4 vegetations. As the reviewer suggests, these could provide explicit estimates of variables for crops vs natural vegetation and for C3 vs C4. In the Discussion, we have suggested analysis based on TRENDY models as a potential avenue for further research.

**Comment 2**

Writing quality and over-generalisation. Although I usually refrain from commenting on style, some issues affect the paper's scientific clarity. I gave a few examples below

Carelessness: L137: "Click or tab here to enter text" is clearly a placeholder. The manuscript contains two "Discussion" sections (4 and 6); Section 6 seems intended as a Conclusion. Please proofread carefully. More examples in minor comments below.

**Response**

Our apologies for these oversights, which have been corrected.

Overstatements: The manuscript repeatedly describes the work as analysing "vegetation diversity" or "vegetation types" (L11, L16, L19, L74, L215, L275). In fact, it focuses on the photosynthetic pathway (C3 vs C4). Terms such as diversity or even vegetation types/PFTs carry broader meaning and could mislead readers. Being precise will not diminish the study's novelty.

**Response**

We agree. We have replaced 'vegetation type' or 'vegetation diversity' with 'photosynthetic pathway' in several places. We have retained 'vegetation type' in some situations where a more general term seems appropriate.

**Referee 2 other Comments and Responses**

L31: "Land Cover and Land Use Change (LUCC)" is a more standard term. Corrected to 'land cover and land use change (LUCC)'.

L38: Molotoks et al.—only one reference is cited; the "a" after 2018 is unnecessary (cf. L286). Corrected.

L43–44: The description of C3 vs C4 lacks detail on anatomical/structural differences that drive their distinct climate responses—important for motivating later arguments.

We have added a statement on the anatomical differences that drive their distinct climate responses.

L70: The "3" in C3 should be subscript. Corrected.

L125: You did not describe how C3/C4 Cveg was estimated; I assume you applied the same method as for GPP. Response: Yes, we applied the same approach. We have revised the text to clarify this point.

L139: Define Δtot before using it.

Done.

L140: Too many simplifying assumptions undermine the credibility of later detailed analyses; the variation in  $\Delta$  is particularly large.

Response: We are not sure what the reviewer meant by this comment. We are estimating the effect of  $C_3/C_4$  vegetation proportion changes alone on global  $\Delta$  – and therefore we have to use a fixed  $\Delta$  for each photosynthetic pathway. We use values based on the Cornwell database.

**L186: Please clarify the source of the 10–45 % cropland figures.**

Response: The percentage of  $C_4$  crop area is calculated as ( $C_4$  area/Total crop area) x 100 for each of the models. The range given is from minimum to maximum for the year 2014. Fig. 4(a) and (g) have been added to the manuscript for improved clarity.

**L201: "UKESM is quite consistent"—please explain.**

This sentence has been restated as 'UKESM1 estimate of the area of natural C4 vegetation for 2000-14 compares well with Luo et al (2024)'s'.

**L224: I remain unconvinced by the C4 GPP and Cveg values.**

We have tried to straightforwardly explain the method we used, and its limitations. We hope that future modelling studies will provide more information about the GPP and carbon content of C3 vs C4 vegetation.

L315: Instead of stating that COS measurements prove models underestimate CO2 fertilisation, consider a more cautious phrasing, e.g. "we noticed there is still difference between models and COS on the quantification of eCO2 effect and highlight the uncertainty in future 13C discrimination rate as eCO2 benefits C3 more than C4…"

Response: This point is noted. However, we find the statement is not overly confident as written, since it says "there may have been": "In particular, since the GPP trend in the models (11 to 18 %, Fig. S8) is weaker than the CO2 fertilization effect over the 20th century based on carbonyl sulphide data (+30 %; Campbell et al. 2017), there may have been a more positive C3 vegetation-driven trend in discrimination in reality than in the models."